# Adipose Tissue-Derived CCL5 Enhances Local Pro-Inflammatory Monocytic MDSCs Accumulation and Inflammation via CCR5 Receptor in High-Fat Diet-Fed Mice

**DOI:** 10.3390/ijms232214226

**Published:** 2022-11-17

**Authors:** Pei-Chi Chan, Chieh-Hua Lu, Hung-Che Chien, Yu-Feng Tian, Po-Shiuan Hsieh

**Affiliations:** 1Department of Physiology & Biophysics, National Defense Medical Center (NDMC), Taipei 114, Taiwan; 2Division of Endocrinology and Metabolism, Department of Internal Medicine, Tri-Service General Hospital, NDMC, Taipei 114, Taiwan; 3Department of Surgery, Chi Mei Medical Center, Tainan 717, Taiwan; 4Graduate Institute of Medical Sciences, NDMC, Taipei 114, Taiwan; 5Department of Medical Research, Tri-Service General Hospital, Taipei 114, Taiwan

**Keywords:** diet-induced obesity, adipose tissue inflammation, CCL5, monocytic myeloid-derived suppressor cells (MDSCs)

## Abstract

The C-C chemokine motif ligand 5 (CCL5) and its receptors have recently been thought to be substantially involved in the development of obesity-associated adipose tissue inflammation and insulin resistance. However, the respective contributions of tissue-derived and myeloid-derived CCL5 to the etiology of obesity-induced adipose tissue inflammation and insulin resistance, and the involvement of monocytic myeloid-derived suppressor cells (MDSCs), remain unclear. This study used CCL5-knockout mice combined with bone marrow transplantation (BMT) and mice with local injections of shCCL5/shCCR5 or CCL5/CCR5 lentivirus into bilateral epididymal white adipose tissue (eWAT). CCL5 gene deletion significantly ameliorated HFD-induced inflammatory reactions in eWAT and protected against the development of obesity and insulin resistance. In addition, tissue (non-hematopoietic) deletion of CCL5 using the BMT method not only ameliorated adipose tissue inflammation by suppressing pro-inflammatory M-MDSC (CD11b^+^Ly6G^−^Ly6C^hi^) accumulation and skewing local M1 macrophage polarization, but also recruited reparative M-MDSCs (CD11b^+^Ly6G^−^Ly6C^low^) and M2 macrophages to the eWAT of HFD-induced obese mice, as shown by flow cytometry. Furthermore, modulation of tissue-derived CCL5/CCR5 expression by local injection of shCCL5/shCCR5 or CCL5/CCR5 lentivirus substantially impacted the distribution of pro-inflammatory and reparative M-MDSCs as well as macrophage polarization in bilateral eWAT. These findings suggest that an obesity-induced increase in adipose tissue CCL5-mediated signaling is crucial in the recruitment of tissue M-MDSCs and their trans-differentiation to tissue pro-inflammatory macrophages, resulting in adipose tissue inflammation and insulin resistance.

## 1. Introduction

Obesity and its comorbidities, such as insulin resistance, type 2 diabetes mellitus (T2DM), cardiovascular diseases (CVD), and nonalcoholic fatty liver disease (NAFLD), are now considered burdens on health care system worldwide [1]. Evidence suggests that chronic immune cell infiltration and inflammation in adipose tissue are crucially involved in obesity-associated systemic inflammation and cardiometabolic abnormalities [2,3]. On the other hand, a high-fat diet (HFD) and leptin have been reported to promote tumor progression by inducing myeloid-derived suppressor cells (MDSCs) [4], which also play intrinsic roles in inhibiting obesity-induced inflammation and insulin resistance [5]. Accordingly, obesity-associated MDSCs have been reported to protect against some of the metabolic dysfunction associated with obesity, concomitant with increasing the rate of tumor progression that is characteristic of obese cancer patients [6]. In addition, individuals with obesity typically have elevated blood and adipose tissue levels of IL-6 [7], TNF-α [8], and prostaglandin E_2_ (PGE_2_) [9], which are major inducers of the differentiation, accumulation, and potency of tumor-induced MDSCs. MDSCs have been shown to participate in the phenotypic switch from the pro-inflammatory M1 subtype to the anti-inflammatory M2 subtype, thereby rebalancing the immune response in transplants toward immune tolerance [10]. Moreover, the adoptive transfer of MDSCs has an excellent beneficial effect on autoimmune disease treatment [11] and chronic kidney diseases [12]. However, since the same constellation of molecules is present in white adipose tissue (WAT) [13], it remains unclear whether the pro-inflammatory environment of adipose tissue may support the induction and accumulation of MDSCs [4]. 

MDSCs are described as a heterogeneous population of granulocytic-MDSCs (G-MDSCs) and monocytic-MDSCs (M-MDSCs) [14,15]. G-MDSCs and M-MDSCs are elevated in *ob/ob* and HFD mice [4]. Accordingly, circulating levels of M-MDSCs are elevated in obese humans [16]. Chinese men with overweight and obesity without diabetes or other complicating metabolic issues have elevated M-MDSCs without changes in the number of monocytes in their blood compared to those in normal-weight men [16]. M-MDSCs are defined as CD11b^+^Ly6G^−^Ly6C^+^ cells. Additionally, high expression of the Ly6C antigen (Ly6C^hi^) is generally associated with a pro-inflammatory state, and low expression (Ly6C^low^) is an indicator of an anti-inflammatory or regenerative state in M-MDSCs [17]. The differential recruitment of these monocyte subsets appears to be crucially controlled by chemokines released from injured tissue. CD11b^+^ macrophages recruited from monocytes have been reported to play dual roles in regulating tissue-destructive and resolution/repair events in obesity-induced adipose tissue inflammation [18]. The dual roles of adipose tissue macrophages (ATMs) are mainly determined by their inherent plasticity caused by the polarization of macrophages toward M1 (pro-inflammatory) or M2 (anti-inflammatory) phenotypes in response to various stimuli in the context of obesity or normal weight. However, the distribution and functional role of M-MDSC subsets and macrophage polarization in WAT during the development of obesity remain largely unknown.

Chemokines are chemoattractive cytokines that are substantially involved in obesity-induced adipose tissue inflammation by recruiting inflammatory immune cells from the blood vessels. C-C chemokine motif ligand 5 (CCL5), which is also known as regulated on activation, normal T-cell expressed and secreted (RANTES), functions as a professional chemoattractant that directs the migration of leukocytes into inflammatory lesions during various pathological processes. For instance, previous studies indicated that CCL5/RANTES and its receptor–CCR5 are associated with T2DM, glucose intolerance, obesity, and atherosclerosis [19,20,21,22]. On the other hand, it has been demonstrated that targeted inhibition of the autocrine CCL5/CCR5 axis could reprogram immunosuppressive myeloid cells and suppress tumor progression [23]. However, it is not known whether the CCL5/CCR5 axis is involved in obesity-driven MDSCs and insulin resistance. The present results suggest that the tissue CCL5/CCR5 signaling pathway is important in the recruitment of M-MDSCs and the differentiation of M-MDSCs to M1/M2 ATMs. This local signaling, especially in adipose tissue, further promotes inflammatory reactions by facilitating pro-inflammatory M-MDSCs (CD11b^+^Ly6G^−^Ly6C^hi^) and M1 macrophage polarization and inhibits reparative M-MDSCs (CD11b^+^Ly6G^−^Ly6C^low^) and M2 macrophage polarization, ultimately contributing to the development of obesity-associated adipose tissue dysfunction and insulin resistance.

## 2. Results 

### 2.1. Mice with CCL5 Gene Deletion Are Protected against HFD-induced Obesity and Insulin Resistance

To determine whether CCL5 is required for obesity-induced adipose tissue inflammation and insulin resistance, we first examined the metabolic phenotype of global CCL5 knockout (CCL5KO) mice. Body weight was not different between WT and CCL5KO mice fed a normal chow diet (NCD). Global deletion of CCL5 attenuated HFD-induced weight gain in mice during the 18-week diet intervention period (Figure 1A). The deletion of CCL5 did not affect the food intake of mice fed a NCD or HFD (Figure 1B). HFD feeding for 18 weeks increased fasting blood glucose and insulin in WT mice, but not in those with CCL5 deletion (Figure 1C,D). Accordingly, HOMA-IR in CCL5KO mice was significantly lower than that in WT mice fed a HFD (Figure 1E). The deposition of adipose tissue lipid and adipocyte inflammation increase the risk of insulin resistance, which in turn exacerbates obesity and metabolic diseases [24]. CT scan analysis showed that subcutaneous fat contents in CCL5KO mice were significantly lower than those in WT mice fed a NCD and visceral fat in CCL5KO mice were significantly smaller than those in WT mice on HFD feeding (Figure 1F,G). These observations indicated that CCL5 deletion could reduce lipid accumulation in adipose tissue and improve glucose intolerance and insulin resistance in HFD-fed mice.

### 2.2. CCL5 Deficiency Ameliorates Adipose Tissue Inflammation in HFD-Fed Mice

We further explored the potential effects of CCL5-mediated adipose tissue inflammation. Adipose tissue inflammation could be, at least in part, attributed to an increase in inflammation-promoting M1 macrophages and a decrease in anti-inflammatory M2 macrophages [25,26,27]. Likewise, compared with that in HFD-fed WT mice, the elevated expression of M1 macrophage–related genes (TNF-α, iNOS, and MCP1) was significantly attenuated in CCL5KO mice, but the decreased expression of the M2 macrophage marker Arg1 in the eWAT of HFD-fed WT mice was abrogated. There were no differences in CD206 and IL-4R gene expressions in these parameters between the WT and CCL5KO mice (Figure 2A). Furthermore, HFD-fed WT mice exhibited significantly increased TNF-α and IL-6 protein levels (Figure 2B,C) and decreased IL-4, IL-10 and IL-13 protein levels in eWAT compared to NCD-fed WT mice (Figure 2D–F). Deletion of CCL5 increased IL-10 (another M2 macrophage marker [28]) protein levels, and suppressed TNF-α and IL-6 protein levels in the eWAT of HFD-fed mice (Figure 2B–F). Accordingly, FACS analysis showed that CCL5 deletion significantly suppressed the HFD-induced increase in the number of M1 macrophages (defined as F4/80^+^CD11c^+^CD206^−^) in the stromal vascular fractions (SVFs) of eWAT compared to those in WT mice (Figure 2G). Conversely, the HFD-induced suppression of alternatively activated anti-inflammatory M2 macrophages (defined as F4/80^+^CD11c^−^CD206^+^) was reversed in CCL5KO mice (Figure 2H). We next sought to examine the role of CCL5 in obesity-associated MDSC accumulation in eWAT. The percentage of circulating G-MDSCs (defined as CD11b^+^Ly6G^+^Ly6C^−^) was significantly lower in CCL5KO mice than in WT mice after NCD and HFD feeding. There was no difference between experimental animals fed the NCD and HFD (Figure 3A,B). On the other hand, the level of M-MDSCs (defined as CD11b^+^Ly6G^−^Ly6C^+^) was progressively increased in CCL5KO mice in response to HFD feeding (Figure 3C,D) but was not changed in WT mice, suggesting that the biogenesis of M-MDSCs may be altered during HFD feeding in CCL5KO mice. CD11b^+^Ly6G^−^Ly6C^hi^ inflammatory M-MDSCs are considered to be precursors of macrophages and dendritic cells during inflammatory conditions, whereas CD11b^+^Ly6G^−^Ly6C^low^ reparative M-MDSCs represent steady-state precursor cells of tissue macrophages [29,30]. The flow cytometric density of CD11b and Ly6C expression in SVFs of eWAT from experimental mice, which are representative of the 12 mice in each group, is shown in Figure 3E. The elevated population of CD11b^+^Ly6G^−^Ly6C^hi^ pro-inflammatory M-MDSCs in the eWAT of HFD-fed mice was significantly attenuated in CCL5KO mice. However, CCL5 deficiency significantly enhanced the population of CD11b^+^Ly6G^−^Ly6C^low^ reparative M-MDSCs (Figure 3F,G). Our results indicated that global CCL5 deletion ameliorated HFD-induced adipose tissue inflammation and insulin-mediated glucose metabolism in mice, which might be attributed to the decreases in pro-inflammatory M-MDSCs and M1 macrophage and increases in reparative M-MDSCs and M2 macrophage infiltration in eWAT. 

### 2.3. Tissue-Specific but Not Bone Marrow (BM)-Specific CCL5 Knockout Mice Are Protected against HFD-Induced Obesity, Adipose Tissue Inflammation, and Insulin Resistance

Global CCL5KO mice were protected against HFD-induced obesity, adipose tissue inflammation, and insulin resistance. However, it is unclear whether the effect of global CCL5 deficiency is due to tissue or myeloid cell induction. Therefore, we compared the mRNA expression of CCL5 and its receptors (CCR1, CCR3, and CCR5) in the BM and eWAT of WT mice fed a NCD or HFD (Appendix A). Elevated mRNA expression levels of CCL5 and CCR5 were observed in the eWAT of HFD-fed WT mice. In contrast, there was no difference in the mRNA expression of CCL5 (Appendix A) or its receptors (Appendix A) in the BM of NCD-fed and HFD-fed WT mice. To more specifically assess whether the effects depended upon tissue CCL5 expression, we generated bone marrow transplant (BMT) chimeras. Real-time PCR analyses of CCL5 mRNA expression and plasma CCL5 protein levels confirmed CCL5 deletion in adipose tissue and the BM of CCL5KO mice (Appendix A). Young WT and CCL5KO mice were irradiated and given the BM of WT mice to generate WT/WT and WT/L5KO (tissue-specific CCL5 knockout) mice. These mice began HFD feeding after recovering from BMT for 6 weeks. After 20 weeks of HFD feeding, the HFD-induced increases in body weight gain observed in WT/WT mice were significantly suppressed in WT/L5KO mice (Figure 4A). Accordingly, the subcutaneous and epididymal fat pad weight to body weight ratios were higher in WT/WT mice than in WT/L5KO mice fed a HFD (Figure 4B). The HFD-induced increases in insulin and HOMA-IR in WT/WT mice were significantly attenuated in WT/L5KO mice, which was no difference in fasting glucose (Figure 4C–E). These results suggest that tissue CCL5 is crucial for improving obesity-associated adipose tissue lipid accumulation and insulin resistance.

To examine the role of myeloid-derived CCL5 in obesity-associated metabolic abnormalities, we used BM-specific BMT chimeras in which WT mice were irradiated and given the BM of WT or CCL5KO mice to generate L5KO/WT and WT/WT mice and examine whether the loss of CCL5 from hematopoietic cells could influence adipose tissue inflammation. There were no significant changes in body weight, percentage of fat mass, fasting glucose levels, insulin, or HOMA-IR in HFD-fed L5KO/WT mice compared with HFD-fed WT/WT mice (Appendix A). In addition, deletion of CCL5 exclusively in hematopoietic cells did not alter circulating G-MDSC and M-MDSC levels (Appendix A). In addition, HFD-induced changes in inflammatory M-MDSCs, reparative M-MDSCs, M1 and M2 macrophages were not different between L5KO/WT and WT/WT mice (Appendix A). HFD-induced increases in the gene expression of TNF-α, MCP1, and IL-6 in eWAT were not changed between the two groups (Appendix A). As shown in Appendix A, the mRNA levels of IL-10 and IL-13 in eWAT were slightly increased in L5KO/WT mice compared to WT/WT mice. HFD feeding decreased IL-10 expression in eWAT but not IL-13 expression in L5KO/WT mice.

### 2.4. Tissue-Specific CCL5 Is Critical for the Recruitment of CD11b^+^Ly6C^low^ M-MDSCs and M2 Macrophages via CCR5 Receptor

Flow cytometric analysis revealed a significant increase in the proportion of circulating M-MDSCs but not G-MDSCs in WT/L5KO mice compared to WT/WT mice fed a NCD and HFD (Figure 5A,B). Furthermore, the proportion of CD11b^+^Ly6G^−^Ly6C^hi^ pro-inflammatory M-MDSCs (Figure 5C), which are preferentially recruited into inflamed tissues and transformed into M1-like macrophages in eWAT, along with M1 macrophages, was substantially increased in WT/WT mice fed a HFD for 20 weeks (Figure 5E). These increases in CD11b^+^Ly6G^−^Ly6C^hi^ pro-inflammatory M-MDSCs and M1 macrophages observed in HFD-fed WT/WT mice were significantly attenuated in WT/L5KO mice. Moreover, HFD-induced suppression of the levels of CD11b^+^Ly6G^−^Ly6C^low^ reparative M-MDSCs and M2 macrophages was reversed in WT/L5KO mice (Figure 5D,F). Accordingly, the increased gene expression of M1-like markers, such as MCP-1 (Figure 5G), TNF-α (Figure 5H), and IL-6 (Figure 5I) in the eWAT of WT/WT mice fed a HFD was significantly attenuated in WT/L5KO mice. In addition, the expressions of M2-related genes such as IL-10 and IL-13, in the eWAT of WT/L5KO mice were significantly increased compared to those in WT/WT mice fed a NCD or HFD (Figure 5J,K). 

To clarify whether the increase in CCL5/CCR5 signaling in eWAT can affect adipose tissue inflammation, lentivirus-derived CCL5 and CCR5 full-length cDNA or CCL5 and CCR5 shRNA were injected directly into bilateral eWAT (Figure 6A). Real-time PCR analyses were used to confirm the efficiency of CCL5/CCR5 overexpression or knockdown in eWAT of experiment mice (Appendix A). After local lentiviral injection for 5 days, the mice were sacrificed, and cells isolated from SVFs of eWAT were subjected to flow cytometric analysis of the subpopulations of M-MDSCs and macrophages (Figure 6B). Flow cytometry analysis indicated a higher proportion of CD11b^+^Ly6G^−^Ly6C^hi^ pro-inflammatory M-MDSCs (Figure 6C) and M1 macrophages (Figure 6D) and a lower proportion of CD11b^+^Ly6G^−^Ly6C^low^ reparative M-MDSCs (Figure 6C) that transformed into M2 macrophages (Figure 6D) in the eWAT of mice overexpressing CCL5/CCR5 in eWAT than in control mice. Conversely, we knocked down CCL5 and CCR5 expression in the eWAT of mice with lentivirus-derived shRNA followed by long-term HFD feeding for 18 weeks. eWAT-specific knockdown of CCL5/CCR5 signaling (Appendix A) abrogated the HFD-induced increase in the gene expressions of M1-like markers, such as TNF-α and IL-6. Slight decreases in the transcript levels of MCP-1 and TNF-α were observed in the eWAT of CCL5/CCR5 double-knockdown mice compared to WT control mice (Figure 6E). On the other hand, the expression of the M2-related gene IL-10 was increased in lentivirus-derived CCL5/CCR5 shRNA-treated HFD-fed mice but not WT mice. There were no differences in IL-13 gene expression in these parameters between mice with CCL5/CCR5 double knockdown and WT control (Figure 6E). Local injection of lentivirus derived-CCL5/CCR5 shRNA promoted the population of CD11b^+^Ly6G^−^Ly6C^low^ reparative M-MDSCs (Figure 6F) and showed no significant changes in M2 macrophages (Figure 6H) and suppressed the population of CD11b^+^Ly6G^−^Ly6C^hi^ inflammatory M-MDSCs (Figure 6G) and M1 macrophages in eWAT of HFD-fed mice (Figure 6I). These data demonstrated that the increase in CCL5-mediated signaling is crucial for facilitating the recruitment and trans-differentiation of CD11b^+^Ly6G^−^Ly6C^hi^ pro-inflammatory M-MDSCs and M1 macrophages and simultaneously suppressing the trans-differentiation of CD11b^+^Ly6G^−^Ly6C^low^ reparative M-MDSCs into M2 macrophages through CCR5 receptor, further deteriorating the adipose tissue inflammatory response in HFD-induced obese mice. 

## 3. Discussion

The importance of chemokines and chemokine receptors in obesity-associated ATM recruitment and insulin resistance has become increasingly evident in recent years [31]. MDSCs have been linked to a wide range of inflammation-associated pathological processes including obesity-associated metabolic disorders [32,33]. Our study demonstrated that tissue (non-hematopoietic) deletion of CCL5, especially in eWAT, not only ameliorate adipose tissue inflammation through suppressing pro-inflammatory M-MDSCs (CD11b^+^Ly6G^−^Ly6C^hi^) accumulation and skewing local M1 macrophage polarization but also recruited reparative M-MDSCs (CD11b^+^Ly6G^−^Ly6C^low^) and insulin-sensitizing M2 macrophages in HFD-induced obese mice. Obesity-induced increases in adipose tissue CCL5-mediated signaling in adipose tissue is crucial for the recruitment of circulating M-MDSCs and trans-differentiation to tissue pro-inflammatory M-MDSCs and macrophages, resulting in adipose tissue inflammation and insulin resistance. 

In the current study, we demonstrated that mice with tissue-derived CCL5 deletion were protected against HFD-induced adipose tissue inflammation and insulin resistance, which was not observed in those with BM-derived CCL5 deletion. It is noteworthy that the metabolic phenotype of tissue-specific CCL5 KO mice was almost identical to that of global CCL5-knockout mice. These results may provide an explanation for the role of CCL5 in the regulation of adipose tissue inflammation during the development of obesity, which is mainly dependent on local CCL5 expression, especially in adipose tissues. Adipose tissue inflammation is closely associated with the development of obesity-associated insulin resistance and other metabolic diseases [24,34]. Previous studies on human adipose tissue showed that the expression and secretion of CCL5 could be promoted by inflammatory stimuli and hypoxia in response to the recruitment and survival of ATMs [20,35]. Increased expression of CCL5 has been shown to mediate the arrest and transmigration of monocytes/macrophages into the damaged site by binding with its receptor CCR5 [36,37,38]. In contrast to transgenic mice models, loss of CCR5 were related to both reduction of total ATMs content and polarization ATMs toward M2 phonotype [39]. Additionally, CCL5 levels were particularly elevated in the stromal vascular fraction of WAT as compared with its adipocyte fraction in HFD-induced obese mice [21]. The present results further provide in vivo evidence supporting the importance of tissue-derived CCL5 in the etiology of obesity-induced adipose tissue inflammation and insulin resistance. 

In the present study, we found that enhanced tissue CCL5 signaling could promote the recruitment of circulating M-MDSCs and tissue CD11b^+^Ly6G^−^Ly6C^hi^ pro-inflammatory M-MDSCs and their subsequent trans-differentiation to M1 macrophages, as well as inflammatory reactions in adipose tissues under obese conditions. Mature and immature myeloid cells include a spectrum between monocytes (defined as M-MDSCs) and neutrophils (defined as G-MDSCs) in cancer and chronic inflammation and are derived from BM and spleen [40], whereas M-MDSCs are characterized by high plasticity. It has been reported that the pro-inflammatory microenvironment in adipose tissue supports the induction and accumulation of MDSCs [5]. Obesity also alters the tumor microenvironment to favor an increase in MDSCs by elevating local CCL2 production in HFD-fed mice [41]. In the tumor microenvironment, it has also been demonstrated that M-MDSCs generated in the BM that migrate to the tumor site are attracted by local CCL5 production [42]. However, deletion of CCL5 in BM precursors was shown to arrest tumor-associated macrophage differentiation and induce robust anti-tumor immunities [23]. The differentiation of M-MDSCs into macrophages is shaped by the tumor microenvironment [43]. As demonstrated previously, MDSCs in obese mice were shown to decrease fasting serum glucose levels and protect against the development of insulin resistance [4,5]. However, the link between adipose tissue-derived CCL5 in the circulation and tissue M-MDSCs in the development of obesity remains elusive. A previous report demonstrated that CD11b^+^Ly6G^−^Ly6C^low^ reparative monocytes were predominantly recruited following acute liver ischemia/reperfusion injury and CD11b^+^Ly6G^−^Ly6C^hi^ inflammatory monocytes were recruited to a lesser extent [30]. M-MDSCs are potent effectors of inflammatory responses and acquired signals in adipose tissue. Our data indicated that the enhancement of adipose tissue CCL5 could direct the accumulation of pro-inflammatory M-MDSCs and the trans-differentiation of tissue M1 macrophages in adipose tissues, resulting in a local inflammatory response and impairing the overall control of glucose homeostasis during the development of obesity.

The recruitment and infiltration of ATMs play a central role in obesity-associated inflammation and related cardiometabolic disorders [44]. As obesity progresses, resident ATMs are mainly converted from anti-inflammatory M2-like macrophages into pro-inflammatory M1-like macrophages [27]. In addition to adipose tissue-resident macrophages, monocyte-derived macrophages which are short-lived cells, are recruited to adipose tissues during inflammation [45]. By utilizing single-cell RNA sequencing technology, Hill et al. identified two discrete ATM populations (CD11b^+^Ly6C^+^ and CD11b^+^Ly6C^−^CD9^+^), which are associated with obesity [46]. CD11b^+^Gr-1 (Ly6G/Ly6C)^+^ MDSCs are another cell type that has the potential to modulate inflammation and is enriched in adipose tissue during obesity [6,47]. These cells were shown to promote the shift in M1-M2 cells toward the M2 phenotype during obesity and subsequently inhibited the progression of type 2 diabetes in obese mice [4]. Our study used loss- and gain-of-function approaches to reveal the role of adipose tissue CCL5 in M-MDSCs subpopulation and trans-differentiation of tissue macrophages in obesity-induced adipose tissue inflammation. Our data suggest that increased tissue CCL5 expression is crucial for increasing CD11b^+^Ly6G^−^Ly6C^hi^ pro-inflammatory M-MDSCs and suppressing CD11b^+^Ly6G^−^Ly6C^low^ reparative M-MDSCs, which may be associated with M1-M2 trans-differentiation and the development of obesity-induced adipose tissue inflammation through CCR5 receptor, which may contribute to the microenvironment that may foster the accumulation of these cells and serve as a feedback mechanism to curb the development of inflammation.

However, although the present study shed light on the dominant role of tissue-derived CCL5 in the metabolic phenotype of HFD-induced obese mice, the impact of myeloid-derived CCL5 could not be completely excluded. For instance, hematopoietic CCL5 has been reported to promote triple-negative mammary tumor progression by regulating the generation of MDSCs [48]. However, the expression of myeloid-derived CCL5 was not significantly changed in HFD-fed mice during the current experimental conditions, indicating the minor role of hematopoietic CCL5 on HFD-induced phenotypic changes in our study. The underlying mechanism by which M-MDSCs trans-differentiation into tissue macrophages in the development of obesity remains unclear and needs to be further examined.

## 4. Materials and Methods

### 4.1. Mouse Models and Tissue Collection

B6.129P2-Ccl5^tm1Hso^/J (CCL5KO) mice (Strain #:005090) were purchased and imported from Jackson Laboratory (Bar Harbor, ME, USA), and the WT littermates were bred and housed in the Animal Center at the National Defense Medical Center, which is certified by AAALAC. The experimental mice were fed an NCD or HFD (D12451, 45% of kcal from fat, 35% of kcal from carbohydrate, and 20% of kcal from protein, Research Diets) ad libitum for 18 weeks. Different white adipose tissue deposits (epididymal, subcutaneous fat pads) were collected separately, weighed, and subsequently divided to provide uniform tissue samples.

### 4.2. Food Intake, Body Weight, and Metabolic Parameters

The mice and food intake were weighed weekly. At the end of the experiments, serum samples were collected. An Accu-check glucometer (Roche Diagnostics, Indianapolis, USA) was used to determine blood glucose levels. Insulin was measured with mouse insulin ELISA kits (Mercodia AB, Uppsala, Sweden) according to the manufacturer’s protocols. The homeostasis model assessment index (HOMA-IR) was calculated as follows: fasting plasma insulin (μU/mL) × fasting plasma glucose (mmol/L)/22.5 [49].

### 4.3. Measurement of Adipocytokines Concentrations 

Tissue TNF-α levels were evaluated using a mouse TNF-α DuoSet ELISA kit (R&D Systems, Minneapolis, USA). The protein levels of IL-6, IL-4, IL-10, and IL-13 were measured with the indicated ELISA kits (BioLegend, San Diego, CA, USA) according to the manufacturer’s instructions.

### 4.4. Bone Marrow Transplantation (BMT)

A single-cell suspension of BM was prepared from the femurs and tibias of WT littermate or CCL5KO mice and was suspended in PBS at a concentration of 5 × 10^7^/mL. Recipient WT littermates and CCL5KO mice were irradiated with two doses of 600 rads each, 4 h apart. The recipient mice were then anesthetized and injected with 1 × 10^7^ BM cells IV into the orbital sinus. 

### 4.5. Whole-Body Fat Composition and Distribution Analysis by Quantitative Computed Tomography 

To visualize whole-body adipose tissue accumulation and distribution, including subcutaneous, visceral, and brown fat, male CCL5KO mice and their WT littermates were fed an NCD or HFD and scanned by whole-body animal micro-computed tomography (CT, Quantum FX microCT Imaging System, Perkin Elmer, MA, USA) [50]. The scanned images were colored as follows: yellow, subcutaneous fat; green, visceral fat; white, muscle. Computer tomography scans were performed at 1 mm intervals × 10 slices. The adipose tissues were three-dimensionally reconstructed, and the quantities of subcutaneous, visceral, and total adipose tissues were compared after normalization to mouse body weights.

### 4.6. Local Injection of shCCL5/shCCR5 or CCL5/CCR5 Lentivirus into the Bilateral eWAT of Mice

Recombinant lentiviruses were produced by co-transfecting HEK 293T cells with the transfer vector (CCL) and helper plasmids (psPAX2 and pMD2. G) using Lipofectamine 2000 transfection reagent. A total of 5 × 10^6^ 293T cells were seeded in 10-cm plates for 24 h before transfection in DMEM with 10% fetal bovine serum (FBS), penicillin (100 IU/mL), and streptomycin (100 mg/mL) in a 5% CO2 incubator. The culture medium was changed one hour before transfection. A total of 64 μg of plasmid DNA, including 16 μg of the envelope plasmid Ppax2, 16 μg of the packing plasmid pMD2.G, and 32 μg of the transfer vector plasmid, were used per dish for transfection. After the lentiviruses were collected, we used the Lentivirus Purification Kit (ABM, G171) to purify the lentiviruses and increase their titers. The viral titer was determined by the qPCR Lentivirus Titer Kit (ABM, LV900). Mice aged 8–12 weeks were used in the experiments. To induce obesity, WT mice were fed an HFD for 18 weeks. For eWAT injections, control, shCCL5/shCCR5 or CCL5/CCR5 lentivirus (1 × 10^10^ ifu/mouse) was injected directly into bilateral eWAT in a total volume of 50 μL.

### 4.7. Flow Cytometry

Epididymal fat pads from experimental mice were minced and digested for 30 min at 37 °C with 1.5 mg/mL of type II collagenase (Sigma-Aldrich, St. Louis, MO, USA) in KRBH buffer (10 mM HEPES (pH 7.4), 15 mM NaHCO3, 120 mM NaCl, 4 mM KH2PO4, 1 mM MgSO4, 1 mM CaCl_2_, and 2 mM sodium pyruvate) containing 5% BSA. SVFs were resuspended in PBS supplemented with 2% FBS and incubated with Fc-Block (BD Bioscience, San Jose, CA, USA), followed by fluorochrome-conjugated antibodies (Appendix A). Cells were analyzed using the FACSCanto™ II System (BD Bioscience, San Jose, CA, USA). Data analysis and compensation were performed using FlowJo.

### 4.8. Immunoblots

Total tissues were homogenized and sonicated in RIPA lysis buffer. The protein extracts were subjected to SDS-polyacrylamide gel electrophoresis and immunoblotting using the following primary antibodies with working dilution 1:1000 for CCL5 (GeneTex, Irvine, CA, USA, GTX 108441), CCR5 (GeneTex, Irvine, CA, USA, GTX 101330), and β-actin (Sigma-Aldrich, St. Louis, MO, USA).

### 4.9. Quantitative Real-Time PCR

Quantitative real-time PCR was performed on a LightCycler 480 system (Roche) using SYBR Green mix (KAPA SYBR FAST qPCR kit Master Mix, KAPA Biosystems, Wilmington, MA, USA) as described previously [51]. Each sample was measured in duplicate. The following conditions were used for amplification: an initial holding stage of 3 min at 95 °C, followed by 45 cycles of denaturation at 95 °C for 3 s and annealing/extension at 60 °C for 30 s. Products were analyzed by performing a melting curve at the end of the PCR. The data were normalized to 18S mRNA expression and relative quantification was performed using the ΔΔCt method. The primers used in this study are listed in Appendix A.

### 4.10. Statistical Analysis

All data are presented as means ± SEM. The sample size was estimated on the basis of the known variability of the assays. All statistics were calculated using Microsoft Excel and GraphPad Prism. Significant differences between two groups were evaluated by unpaired Student’s *t*-test (for comparison of only 2 groups) or one-way ANOVA (for comparison more than three groups) followed by a Bonferroni’s post hoc test. *p* < 0.05 was adopted as significant.

## 5. Conclusions

This study demonstrated that enhanced tissue CCL5-mediated signaling, especially in adipose tissues, is crucial in the development of circulating and tissue M-MDSCs and the trans-differentiation of tissue macrophages, ultimately resulting in adipose tissue inflammation and insulin resistance via CCR5 receptor. These results suggest that targeting adipose tissue CCL5/CCR5-mediated signaling could have therapeutic potential for controlling the development of obesity-associated adipose tissue inflammation and metabolic abnormalities. 

## Figures and Tables

**Figure 1 ijms-23-14226-f001:**
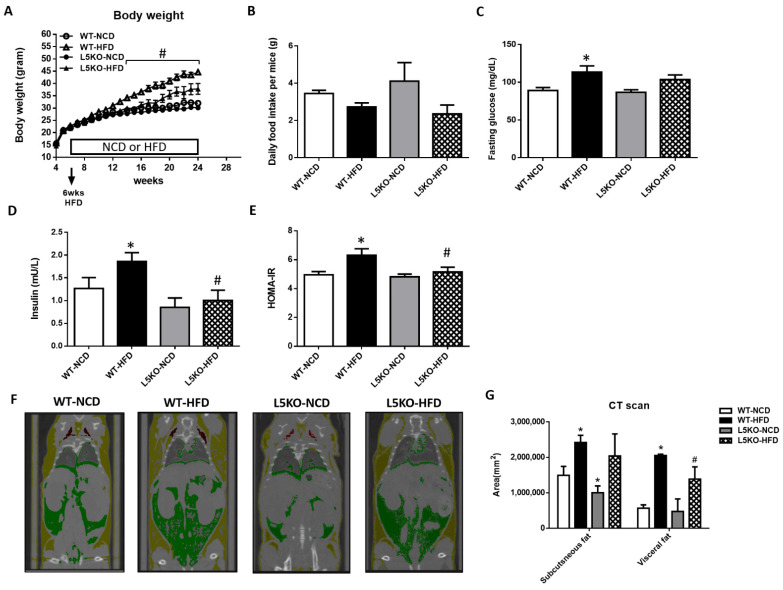
Protection against HFD-induced obesity in CCL5 knockout (CCL5KO) mice. WT and CCL5KO mice were fed high fat diet (HFD) or normal chow diet (NCD) (*n* = 9–12 per group) for 18 weeks beginning at 6 weeks of age. (**A**) Body weight; (**B**) daily food intake; (**C**) fasting blood glucose; (**D**) insulin; (**E**) HOMA-IR; (**F**) representative microCT images of defined objects of bone, soft tissue, visceral fat, and subcutaneous fat based on density thresholds; (**G**) quantification of adipose volume based on defined objects within microCT scans. All data are presented as mean values ± SEM from at least 9 mice in each group. Statistical differences are indicated: *, *p* < 0.05 vs. WT-NCD; #, *p* < 0.05 vs. WT-HFD.

**Figure 2 ijms-23-14226-f002:**
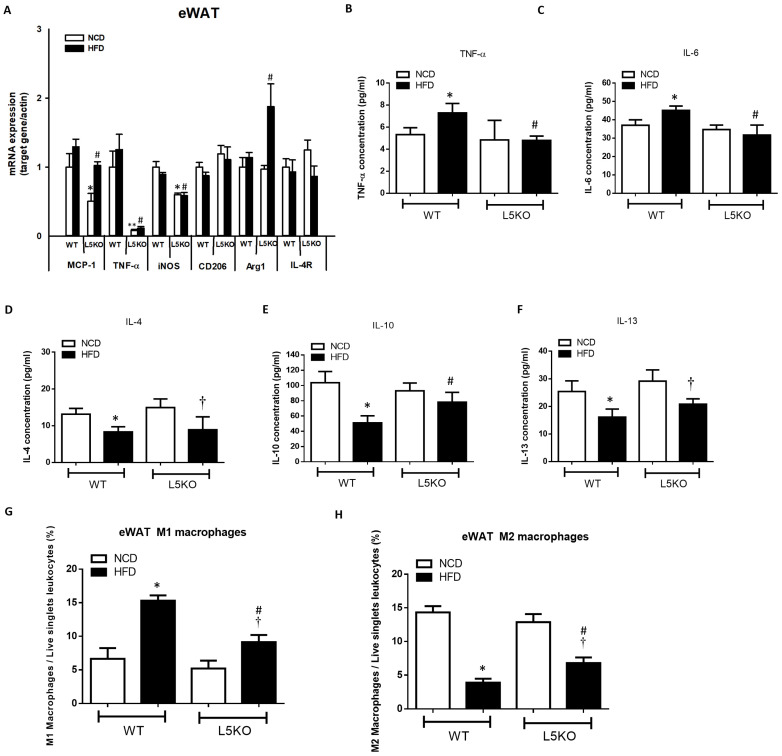
CCL5 deficiency ameliorates M1 macrophage infiltration and inflammation in the eWAT of HFD-fed mice. (**A**) Gene expressions of the M1 macrophage markers MCP-1, TNFα, and iNOS, and the M2 macrophage markers CD206, Arg1, and IL-4R using total mRNA extracted from eWAT and qPCR analysis; adipose tissue protein content of (**B**) TNF-α, (**C**) IL-6, (**D**) IL-4, (**E**) IL-10, and (**F**) IL-13; (**G**) analysis of F4/80^+^CD11c^+^CD206^−^ M1 macrophages in the SVCs of eWAT by flow cytometry; (**H**) analysis of F4/80^+^CD11c^−^CD206^+^ M2 macrophages in the SVCs of eWAT by flow cytometry in CCL5KO and WT mice fed an NCD or HFD for 18 weeks. All data are presented as mean values ± SEM from at least 9 mice in each group. Statistical differences are indicated: *, *p* < 0.05; **, *p* < 0.001 vs. WT-NCD; #, *p* < 0.05 vs. WT-HFD; †, *p* < 0.05 vs. L5KO-NCD.

**Figure 3 ijms-23-14226-f003:**
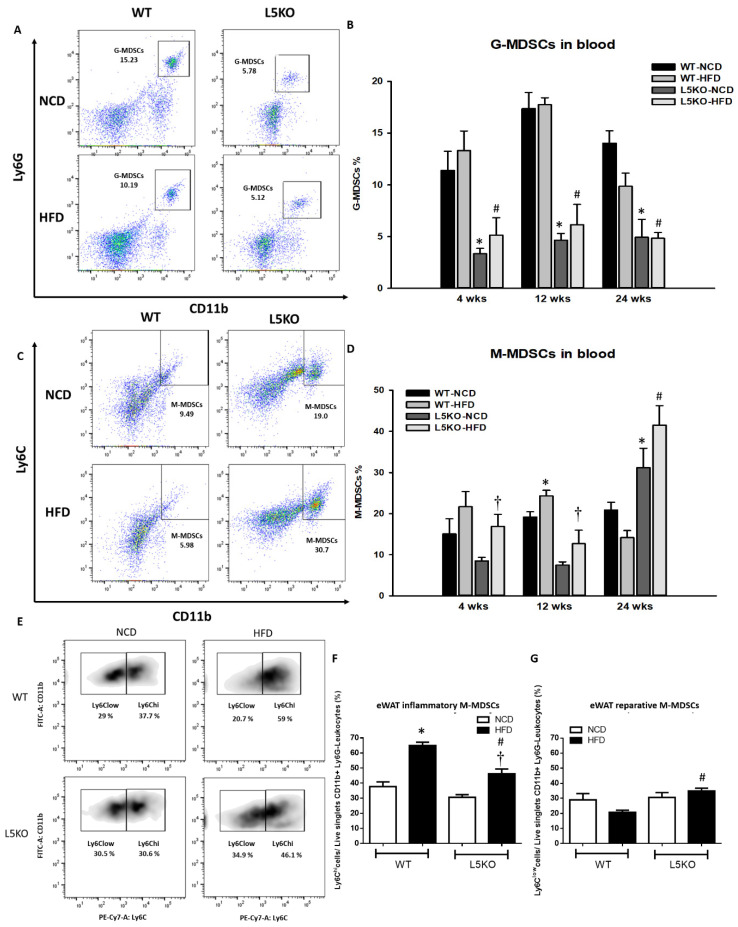
Recruitment of MDSCs into obesity-induced inflamed eWAT. Circulating (**A**,**B**) G-MDSCs and (**C**,**D**) M-MDSCs in WT and CCL5KO mice at 4, 12, and 24 weeks of age. Isolated SVF cells from eWAT were stained for CD11b, Ly6C, and Ly6G, and analyzed by flow cytometry. (**E**,**F**) The CD11b^+^Ly6G^−^Ly6C^hi^ population represents pro-inflammatory M-MDSCs, and (**E**,**G**) the CD11b^+^Ly6G^−^Ly6C^low^ subset represents reparative M-MDSCs in the SVCs of the eWAT via flow cytometry in WT and CCL5KO mice fed NCD and HFD diet for 18 weeks. Flow cytometric plots are representative of 12 mice/group of each group and data are presented as means ± SEM. Statistical difference is indicated: *, *p* < 0.05 vs. WT-NCD; #, *p* < 0.05 vs. WT-HFD; †, *p* < 0.05 vs. L5KO-NCD.

**Figure 4 ijms-23-14226-f004:**
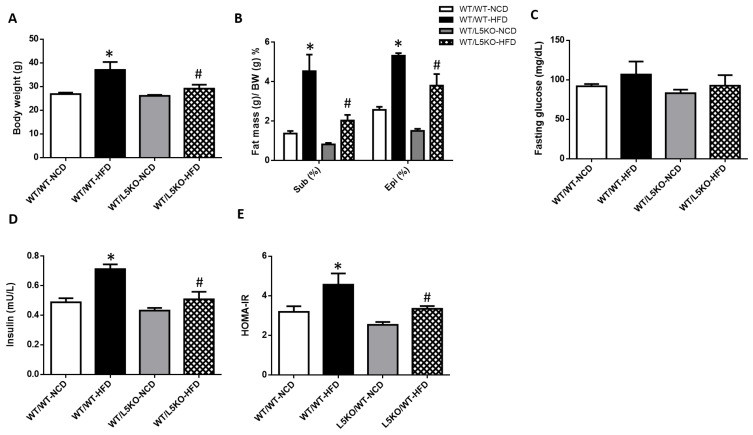
Tissue-specific CCL5 deletion protects against HFD-induced obesity and insulin resistance. WT/WT and WT/L5KO (tissue-specific CCL5 knockout) mice were fed an HFD or NCD (*n* = 6–8 per group) for 20 weeks. (**A**) Body weight; (**B**) subcutaneous and epididymal fat mass per gram body weight; (**C**) fasting blood glucose; (**D**) insulin; (**E**) HOMA-IR. All data are presented as mean values ± SEM from at least 6 mice in each group. Statistical differences are indicated: *, *p* < 0.05 vs. WT/WT-NCD #, *p* < 0.05 vs. WT/WT-HFD.

**Figure 5 ijms-23-14226-f005:**
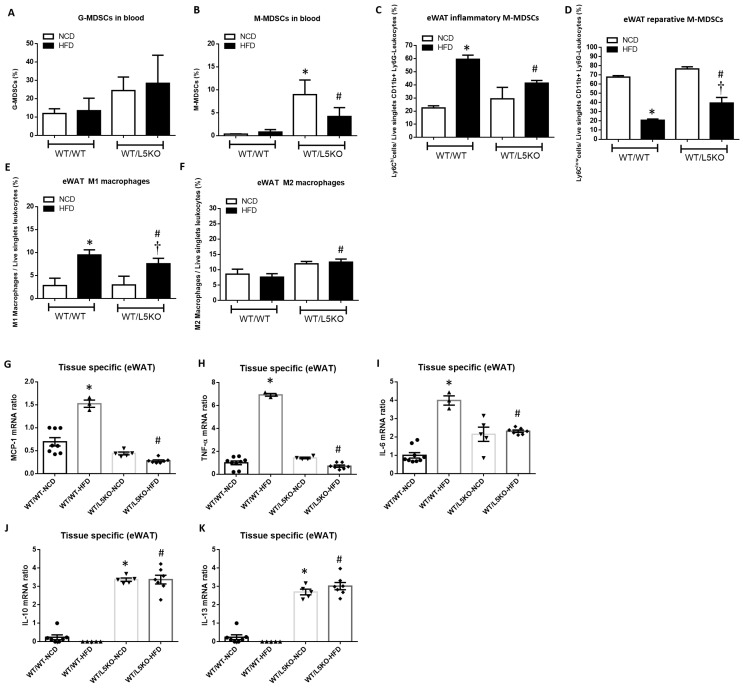
Tissue-specific CCL5 deletion ameliorates M1 macrophage infiltration and recruits MDSCs into obesity-induced inflamed eWAT. (**A**) Circulating G-MDSCs and (**B**) M-MDSCs in WT/WT and WT/L5KO mice fed an NCD or HFD diet for 20 weeks. Isolated SVF cells from eWAT were stained for CD11b, Ly6C, and Ly6G and analyzed by flow cytometry. (**C**) The CD11b^+^Ly6G^−^Ly6C^hi^ population represents pro-inflammatory M-MDSCs, and (**D**) the CD11b^+^Ly6G^−^Ly6C^low^ subset represents reparative M-MDSCs. (**E**) Analysis of F4/80^+^CD11c^+^CD206^−^ M1 macrophages and (**F**) F4/80^+^CD11c^-^CD206^+^ M2 macrophages in the SVCs of eWAT via flow cytometry. Adipose tissue protein levels of (**G**) MCP-1, (**H**) TNF-α, (**I**) IL-6, (**J**) IL-10, and (**K**) IL-13 in WT/WT and WT/L5KO mice fed an NCD or HFD diet for 20 weeks. All data are presented as mean values ± SEM from at least 6 mice in each group. Statistical differences are indicated: *, *p* < 0.05 vs. WT/WT-NCD; #, *p* < 0.05 vs. WT/WT-HFD; †, *p* < 0.05 vs. WT/L5KO-NCD.

**Figure 6 ijms-23-14226-f006:**
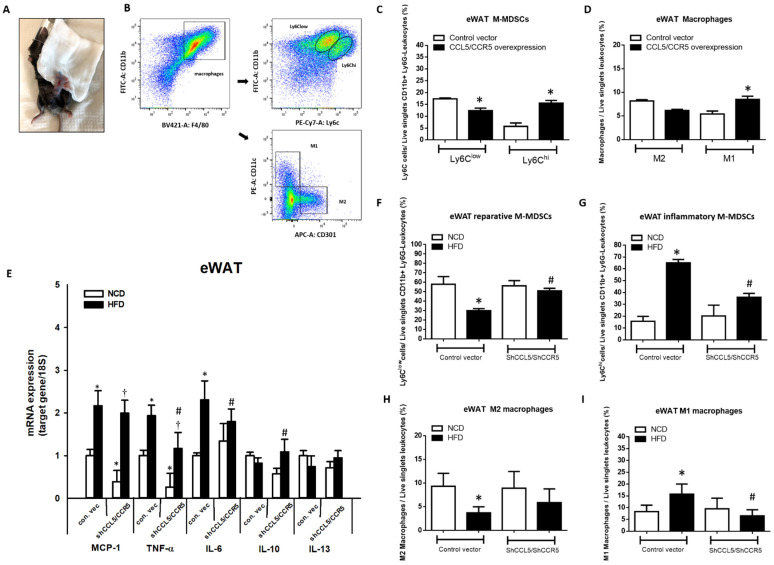
Overexpression and knockdown of local CCL5/CCR5 expression substantially affected the recruitment and trans-differentiation of pro-inflammatory M-MDSCs and M1 macrophages in bilateral eWAT of WT mice. (**A**) Mice were injected with either control and lentivirus-derived CCL5 and CCR5 full-length cDNA or CCL5 and CCR5 shRNA (*n* = 5 each group) into bilateral eWAT; the injected tissues were harvested on day 5 post-injection to analyze CCL5/CCR5-mediated local effects via flow cytometry; (**B**) flow cytometric analysis of (**C**) the M-MDSC population and (**D**) M1 and M2 macrophages in the SVCs of eWAT injected with lentivirus-derived CCL5 and CCR5 full-length cDNA; (**E**) total mRNA was extracted from eWAT injected with CCL5 and CCR5 shRNA and used for qPCR analysis of the gene expression of M1 macrophage markers and M2 macrophage markers; (**F**) the CD11b^+^Ly6G^−^Ly6C^low^ subset represents reparative M-MDSCs; (**G**) the CD11b^+^Ly6G^−^ Ly6C^hi^ population represents pro-inflammatory M-MDSCs; (**H**) M2 macrophages; (**I**) M1 macrophages in the SVCs of treated eWAT. All data are presented as mean values ± SEM from at least 5 mice in each group. Statistical differences are indicated: *, *p* < 0.05 vs. control-vector NCD; #, *p* < 0.05 vs. control-vector HFD; †, *p* < 0.05 vs. shCCL5/shCCR5-NCD.

## Data Availability

Not applicable.

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
