# Peer review of "Adipose Tissue-Derived CCL5 Enhances Local Pro-Inflammatory Monocytic MDSCs Accumulation and Inflammation via CCR5 Receptor in High-Fat Diet-Fed Mice"

_ijms, 2022, doi:10.3390/ijms232214226_

Round 1

Reviewer 1 Report

This study evaluates, in a mouse model, the role of the chemokine CCL5 in the inflammation of adipose tissue induced by a high fat diet resulting in increased insulin resistance. The authors first compared the level of weight gain and adipose tissue quantity in either wild type mice or mice with a global CCL5-KO purchased from Jackson lab fed a normal diet or a HFD. They also evaluated the level of monocyte-derived MDSC, their pro-inflammatory or reparative profile and their ability to generate M1 or M2 macrophages within adipose tissue. To try to decipher whether CCL5, which was involved into adipose tissue modifications, originated from bone marrow or other tissues MDSC, they irradiated mice and transplanted them with either WT or CCL5-KO bone marrow. They also further evaluated the role of CCL5 and its receptor CCR5 in epididymal fat by overexpressing CCL5/CCR5 through lentivirus or decreasing their expression through shCCL5/shCCR5 lentiviruses injected into epididymal WAT. All these experiments represent a huge amount of work and manipulations giving rise to a number of results. While the general message related to the role of CCL5 in adipose tissue inflammation and in insulin resistance and in M-MDSC and macrophage phenotype is relevant, a number of controls are missing. Therefore, it is mandatory that this study is completed. The authors need to give access to a number of controls/verifications in particular for WB (see below) and for those regarding the efficiency of the manipulations of the level of CCL5 and CCR5.

Major comments

In Fig 1, in L5KO-NCD mice, there is a major decrease in SCAT and VAT as compared to the WT-NCD control. This is unexpected since the global weight is not decreased (Fig 1A). How do the authors explain this discrepancy?

In fig 2, the authors show Western blots. First, the good controls for phosphorylated forms of the enzyme/proteins are missing. The control is the non-phosphorylated form of the same protein not actin. Second, in what the authors call “graphical data for figure 2A”, they present WB data that are different from those in fig 2. There are two bands for JNK which is expected since there are JNK1 and JNK2, but this is not seen in fig 2. Conversely, there are two bands for p-IKBalpha which is not expected. The histograms do not reflect the levels of the bands which are shown (for example the level of p-IKBalpha WT-NCD and WT-HFD seem to be similar in the graphical abstract figure while there is a significant difference in the corresponding histogram). The authors indicate that they tested 9 mice in each group, but less WB rows are shown.

The authors performed several experiments to KO CCL5 or CCL5 and CCR5 in bone marrow, in adipose tissue or other tissues. They have to show that their deletion was effective by measuring the mRNA level of these factors.

Moreover, they used lentiviruses injected into eWAT to overexpress CCL5/CCR5: they have to show if the overexpression was homogeneous in eWAT, and also the level of the corresponding mRNA

Regarding the sh experiments, animals were fed for 18 weeks after lentivirus injection. It is mandatory to show whether after this time the expression of CCL5 and CCR5 was reduced in eWAT and again whether this reduction was homogeneous. In fig 6H and I, the difference between SHNCD and SHHFD seems not significant (large error bars). Please check.

The authors have missed an important previous study on CCR5 KO which needs to be added and discussed (H Kitade et al Diabetes 2012, 61:1680-1690)

Minor comments

The authors only present data on eWAT. Why not iWAT?

English needs revision

Give the meaning of MDSC in the abstract

Replace ref 19 which do not refer to M-MDSC

References 21 and 37 are the same (I have not checked the other references) and unify the style for bibliography (again ref 21 and 37)

In several figures, the size of the legends in the figure is too small

Author Response

Responses to reviewer #1

(1) In Fig 1, in L5KO-NCD mice, there is a major decrease in SCAT and VAT as compared to the WT-NCD control. This is unexpected since the global weight is not decreased (Fig 1A). How do the authors explain this discrepancy?

Answer: Thanks for the comment from the reviewer. We cautiously recheck the data and number of experimental animals per group in Figure 1G and Figure 1A and redo the statistical analysis. Accordingly, Figure 1G has been revised. In general, the areas (mm2) of SCAT were slightly lower in L5KO-NCD mice than those in WT-NCD mice. There was no difference in the areas of VAT between these two groups. Since total fat mass contributes to around 15% of total body mass in NCD-fed mice (Diabetes. 2010 Jul;59(7):1657-66. doi: 10.2337/db09-1582), the mild increase in the area of subcutaneous fat mass might cause a very slight increase in body weight of L5KO-NCD mice as shown in Figure 1A.

(2) In figure 2, the authors show Western blots. First, the good controls for phosphorylated forms of the enzyme/proteins are missing. The control is the non-phosphorylated form of the same protein not actin. Second, in what the authors call “graphical data for figure 2A”, they present WB data that are different from those in fig 2. There are two bands for JNK which is expected since there are JNK1 and JNK2, but this is not seen in fig 2. Conversely, there are two bands for p-IKBalpha which is not expected. The histograms do not reflect the levels of the bands which are shown (for example the level of p-IKB alpha WT-NCD and WT-HFD seem to be similar in the graphical abstract figure while there is a significant difference in the corresponding histogram). The authors indicate that they tested 9 mice in each group, but less WB rows are shown.

Answer: Thanks for the comment. Due to the limitation of tissue samples and the absence of good control (non-phosphorylated form), Fig 2A-D has been deleted. However, the results from Figure 2E-L still provided solid evidence to support that CCL5 deficiency ameliorates M1 macrophage infiltration and inflammation in eWAT of HFD-fed mice. On the other hand, the total test numbers in Figure 2 in each group were 9-12 mice same as Figure 1. However, due to the protein volume or sample quality of eWAT in each test mouse was not sufficient for all of the measured items, the adipose tissues from 5 animals per group were selected for western blot analysis.

(3) The authors performed several experiments to KO CCL5 or CCL5 and CCR5 in bone marrow, in adipose tissue or other tissues. They have to show that their deletion was effective by measuring the mRNA level of these factors.

Answer: Thanks for the reviewer’s comment. Accordingly, we need to further explain how to validate the CCL5 effect in three different animal models used in this study to clarify the role of local CCL5 in obesity-related AT inflammation.

At first, in the experiment conducted with WT mice fed HFD or a normal chow diet, we did measure the mRNA levels of CCL5 and CCR5 from BM and adipose tissue as shown in Supplemental Figure 1. The data showed increases in CCL5 and CCR5 expressions in adipose tissue but not in BM of HFD-fed mice.

Next, the study was conducted with global CCL5 KO mice and BMT method to further evaluate the effect of BM or tissue-derived CCL5 on the metabolic phenotype of HFD-fed mice. Before the BMT experiment, we have confirmed the genotype of CCL5 KO mice (Strain #:005090, purchased from Jackson Laboratory) which did not express CCL5 in BM and adipose tissue as shown in Supplemental Figure 2 as described previously (Diabetologia. 2011 Apr;54(4):888-99. doi: 10.1007/s00125-010-2020-5). CCL5 KO mice have served as donor mice provided BM-derived cells to test tissue-derived CCL5 effect as well as recipient mice to test BM-derived CCL5 effect.       

To further verify the importance of adipose tissue CCL5 and CCR5 signaling in mediating obesity-induced adipose tissue inflammation in mice, we employed a lentiviral vector to knock down or overexpress CCL5/CCR5 in bilateral eWAT and the expressions of CCL5/CCR5 mRNA and protein levels in eWAT in this part experiment has been added in the Supplemental Figure 5. 

(4) Moreover, they used lentiviruses injected into eWAT to overexpress CCL5/CCR5: they have to show if the overexpression was homogeneous in eWAT, and also the level of the corresponding mRNA

Answer:  In this part experiment, the result of CCL5/CCR5 mRNA and protein levels in eWAT has been added in Supplemental Figure 5A-B. Accordingly, the relevant statements of this study have been added to the results 2.4 in line 261-263, page 9.

(5) Regarding the sh experiments, animals were fed for 18 weeks after lentivirus injection. It is mandatory to show whether after this time the expression of CCL5 and CCR5 was reduced in eWAT and again whether this reduction was homogeneous. In fig 6H and I, the difference between SHNCD and SHHFD seems not significant (large error bars). Please check.

Answer: The result of CCL5/CCR5 mRNA and protein levels in eWAT at the end of the experiment has been added in Supplemental Figure 5C-D. On the other hand, we recheck the data and number of experimental animals per group in Figure 6H and I, and redo cautiously the statistical analysis. The Figure 6H and I have been revised accordingly.

(6) The authors have missed an important previous study on CCR5 KO which needs to be added and discussed (H Kitade et al Diabetes 2012, 61:1680-1690)

Answer: The relevant statements about the previous study on CCR5 KO mice have been addressed to the discussion as suggested in line 331-332, page 12.

Minor comments

(1) The authors only present data on eWAT. Why not iWAT?

Answer: The epididymal WAT represents visceral fat and is more metabolically and inflammatory active than iWAT. The lipid accumulation and inflammatory response of visceral adipose tissues have been speculated to play the central role in the development of obesity-associated metabolic disorders (Obes Rev. 2010 Jan;11(1):11-8. doi: 10.1111).

(2) English needs revision

Answer: The manuscript has been submitted to Springer Nature Author Services for Language Editing (Editing certificate: 86F1-996D-3365-4DA9-B4BD) and the grammar errors suggested by the reviewers have also been corrected.

(3) Give the meaning of MDSC in the abstract

Answer: Revised as suggested. Thanks for the comments.

(4) Replace ref 19 which do not refer to M-MDSC

Answer: The related references have been revised as suggested.

(5) References 21 and 37 are the same (I have not checked the other references) and unify the style for bibliography (again ref 21 and 37)

Answer: We have re-checked all the references and revised them as suggested.

(6) In several figures, the size of the legends in the figure is too small

Answer: We have re-checked all the figure legends and the legends in Figure 3A,3C,3E and 6B have been revised as suggested.

Reviewer 2 Report

This manuscript contains extensive amount of in vivo data describing how adipose-derived CCL5 contributes to the development of high fat-diet induced inflammation in white adipose tissue. The manuscript presents clinically important data and is well written. The Authors also provide mechanistic data to underlie the functional importance of the described observations.

Specific suggestions:

1.       L20: MDSCs should be defined in the Abstract.

2.       Due to the current preference in scientific literature of a non-stigmatizing language to describe diseases, adjectives should be avoided. Therefore, e.g. instead of "obese individuals", "individuals with obesity" should be written. Please, amend this in lanes 46 and 60.

3.       In Figure 2A, molecular weight markers should be provided.

4.       Figure 2E: The data of CD206 and IL-4R expression is not discussed in the manuscript text.

5.       The data shown in Figures 2H and J is not discussed in the manuscript text.

6.       Subfigures 2A-B and 2C-D can be combined.

7.       Figure 6E: The data of IL-13 expression is not discussed in the manuscript text.

8.       4.8. The catalogue numbers and working dilutions of the antibodies should be included.

9.       4.10. Paired or unpaired t-test was applied? How the normal distribution of the data was assessed?

10.   Abbreviations should be written in lanes 330, 371, and 404.

11.   Grammatical or spelling errors should be corrected in lanes 23, 52, 95, 183, and 317.

12.   The manuscript lacks a Conclusion section.

Author Response

Responses to reviewer #2

Specific suggestions:

  1. L20: MDSCs should be defined in the Abstract.

Answer: Revised as suggested. Thanks for the comments.

  1. Due to the current preference in scientific literature of a non-stigmatizing language to describe diseases, adjectives should be avoided. Therefore, e.g. instead of "obese individuals", "individuals with obesity" should be written. Please, amend this in lanes 46 and 60.

Answer: Revised as suggested.

  1. In Figure 2A, molecular weight markers should be provided.

Answer: Thanks for the comment. Due to the absence of good control (non-phosphorylated form) as suggested by reviewer 1, Fig 2A-D has been deleted.

  1. Figure 2E: The data of CD206 and IL-4R expression is not discussed in the manuscript text.

Answer: Added as suggested in Line 128-130, page 4.

  1. The data shown in Figures 2H and J is not discussed in the manuscript text.

Answer: Revised as suggested in Line 131-132, page 4.

  1. Subfigures 2A-B and 2C-D can be combined.

Answer: The figure 2 A-D has been deleted for the reason described in question #3.

  1. Figure 6E: The data of IL-13 expression is not discussed in the manuscript text.

Answer: Revised as suggested in Line 276-279, page 10.

  1. 4.8. The catalogue numbers and working dilutions of the antibodies should be included.

Answer: Due to the absence of good control (non-phosphorylated form), Fig 2A-D (WB data) as well as the related statement in method section has been deleted. Nevertheless, we have added the catalogue and working dilution of the antibody using in WB data shown in Supplemental Figure 5 in Materials and Methods section.

  1. 4.10. Paired or unpaired t-test was applied? How the normal distribution of the data was assessed?

Answer: Thanks for the comment from the reviewer. We cautiously re-evaluate the statistical method using in this study. We found the t test is not a suitable statistical analysis for statistical comparisons among 4 groups. Therefore, multiple comparisons among groups in this study were re-calculated by one-way ANOVA with Bonferroni’s post hoc test, which has considered adequate control of the Type I error required to hold the normality assumption. The relevant results and figures have been revised accordingly.

  1. Abbreviations should be written in lanes 330, 371, and 404.

Answer: We have re-checked all the abbreviations and revised them as suggested.

  1. Grammatical or spelling errors should be corrected in lanes 23, 52, 95, 183, and 317.

Answer: Revised as suggested.

  1. The manuscript lacks a Conclusion section.

Answer: The conclusion section has been added as suggested.

Round 2

Reviewer 1 Report

In the revised version the authors have answered most of my criticisms.

They have performed again statistical analysis of their results and now report different results with differences in the statistical analyses. Regarding my major concerns with WB experiments, they have decided to withdraw all these results.

As required, they added controls on the levels of CCL5/CCR5 in overexpression and knock-down experiments. This point is OK

English has been markedly improved.

The manuscript needs to be improved regarding two points.

In the revised version they have left old and new versions of the figures which is confusing.

In the references, there is a mix of references with given name then last name in entire and of references with last name and the initial of the given name. Thus, and I have not checked all the references, ref 16 and 17 is the same. I strongly suggest to use a reference software such as Endnote to correct the references.

Author Response

Thanks for the comments.

(1) We've deleted the old figures. Please see the following attachment.

(2)  We've re-checked all the references and revised them by Endnote. 
